# "When you're hurt and you need serious help you call 999." Educating children about emergency services and appropriate use of 999: An evaluation study of the Blue Light Hub app

Amy L Paine ,[1] Fiona Maclean[2]

¹Cardiff University Centre for Human Developmental Science, School of Psychology, Cardiff University, Cardiff, UK
²Welsh Ambulance Services University NHS Trust, Wales, UK

**Correspondence to**
Dr Amy L Paine;
paineal@cardiff.ac.uk

## ABSTRACT

**Objectives** In the face of unprecedented demand, the Welsh Ambulance Services University NHS Trust developed 'Blue Light Hub': a new app to educate primary school-aged children about emergency services. Our overarching aim was to examine the effectiveness of the app.

**Design** Primary school-aged children from three schools in South Wales, UK, played with the app for 2 hours over 2 weeks in class time. Children completed quizzes to assess their knowledge and awareness of, and confidence in engaging with, emergency services before and after using the app.

**Participants** Our evaluation focused on N=393 children who completed both the pre-test and post-test quizzes. On average, children were 8–9 years old (median school year, Year 4); 47.8% were male and 50.9% were female.

**Results** After using the app, there was a significant increase in the proportion of children who knew of appropriate actions to take in non-emergency scenarios, $\chi^2(1) = 26.01$, and could provide a question a call handler would ask them if they called 999, $\chi^2(1) = 13.79$. There was also an increase in the proportion of children who could identify an National Health Service (NHS) service that could help them if they were unwell, $\chi^2(1) = 33.31$, name different roles in the NHS, $\chi^2(1) = 12.80$ and knew how dialling 111 could help them $\chi^2(1) = 90.05$ (all p values<0.001).

**Conclusion** To our knowledge, Blue Light Hub is the first app of its kind designed to educate primary school-aged children about emergency services. Our findings provide preliminary evidence that the app supports children's knowledge and awareness of emergency services.

## INTRODUCTION

The Welsh Ambulance Services University NHS Trust handles more than 1800 calls a day from across the country. Yet of the 470 653 incidents recorded by the service in 2021, nearly a quarter were judged as non-essential and non-urgent.[1] Some reasons for non-essential engagement with emergency services may include misjudgement of emergency scenarios and perception of risk, a lack

## STRENGTHS AND LIMITATIONS OF THIS STUDY

⇒ A key strength of this study is the large sample of participants who engaged with the Blue Light Hub app during school time.

⇒ Information regarding how teachers implemented the app with pupils (such as group size, level of teacher support) was not available.

⇒ This app was developed during the social restrictions brought about by the COVID-19 pandemic; future apps should be co-produced with children, parents, and teachers.

of awareness of other healthcare services, and a perception of convenience.[2–4]

In the face of unprecedented demand for the service, a key priority of the Welsh Ambulance Services University NHS Trust is to find avenues to maintain engagement with schools and influence the school curriculum to educate pupils about the Trust's services and ways to use health services appropriately.[1] Immersive and interactive experiences are an effective way to educate and familiarise children with National Health Service (NHS) services and reduce anxieties about engaging with them, by giving children and young people opportunities to talk to NHS staff, handle equipment and observe what might happen in different medical scenarios.[5]

However, in-person interactive initiatives are resource and time-intensive and have limited opportunities for repeated exposure. Interactive apps are a useful and popular tool that is increasingly being integrated into classrooms.[6] Evidence suggests that interactive apps are a promising tool for supporting academic development[7] and for addressing public health challenges[8] such as health education.[9] By combining play with technology, interactive apps are low cost, easily

distributed and have the potential to actively engage children with material via feedback and rewards delivered through gameplay and repetition to reinforce knowledge.[7 10]

The Blue Light Hub app was designed by the Welsh Ambulance Services University NHS Trust to educate 7–11 years old about emergency services. The key objectives in the design of the interactive app were to teach children to recognise emergency from non-emergency scenarios and awareness of other services available in non-emergency situations (ie, 111), about what happens when they call 999, the appropriate use of 999, how ambulance resources are dispatched and managed and the different uniforms they might encounter on their NHS journey. To the best of our knowledge, there is currently no other interactive app that aims to support children's engagement with emergency services. In this evaluation, we examined the following questions: Does the use of the Blue Light Hub app increase children's (1) knowledge of, (2) awareness about and (3) confidence in using emergency services? These questions were investigated in the context of the Blue Light Hub app being used by primary school-aged children for 2 weeks in the classroom.

## METHOD
### Participants
Three primary schools in South Wales were invited to participate in the Blue Light Hub app evaluation. For inclusivity, all children were invited to engage with the app and complete the quizzes. Consent was collected from parents and caregivers for the use of their child's data for the study. Our analyses focus on N=393 children who completed both the pre-test and post-test quizzes (n=115 from School A, n=154 from School B and n=124 from School C). On average, children were 8–9 years old (median Year 4; IQR Year 3 (7–8 years old) – Year 5 (9–10 years old)), n=188 (47.8%) were male, n=200 (50.9%) were female (information on gender was missing for 5 participants). Sociodemographic information about the schools was obtained from their most recent school inspection (Estyn) reports. At School A, approximately 37% of pupils were eligible for free school meals in their 2017 report, and at Schools B and C nearly 10% and 8% of pupils were eligible for free school meals, according to their 2016 and 2019 reports, respectively (rolling 3-year national average in 2021 for pupils of statutory school age who are eligible for free school meals in primary schools in Wales is 21%[11]).

### Data collection
The Blue Light Hub app is a fully bilingual (English-Welsh) gaming app designed by Welsh Ambulance Services University NHS Trust and developed by DotC Studios.[12] The app comprises four games (see figure 1). After completing each game, children receive points that equate to the money they can spend in the app's shop to buy accessories for their customisable avatar.

The Blue Light Hub app evaluation took place over 2 weeks in June 2022. Teachers were provided with written instructions for taking part in the evaluation. The instructions detailed that children were to be informed they

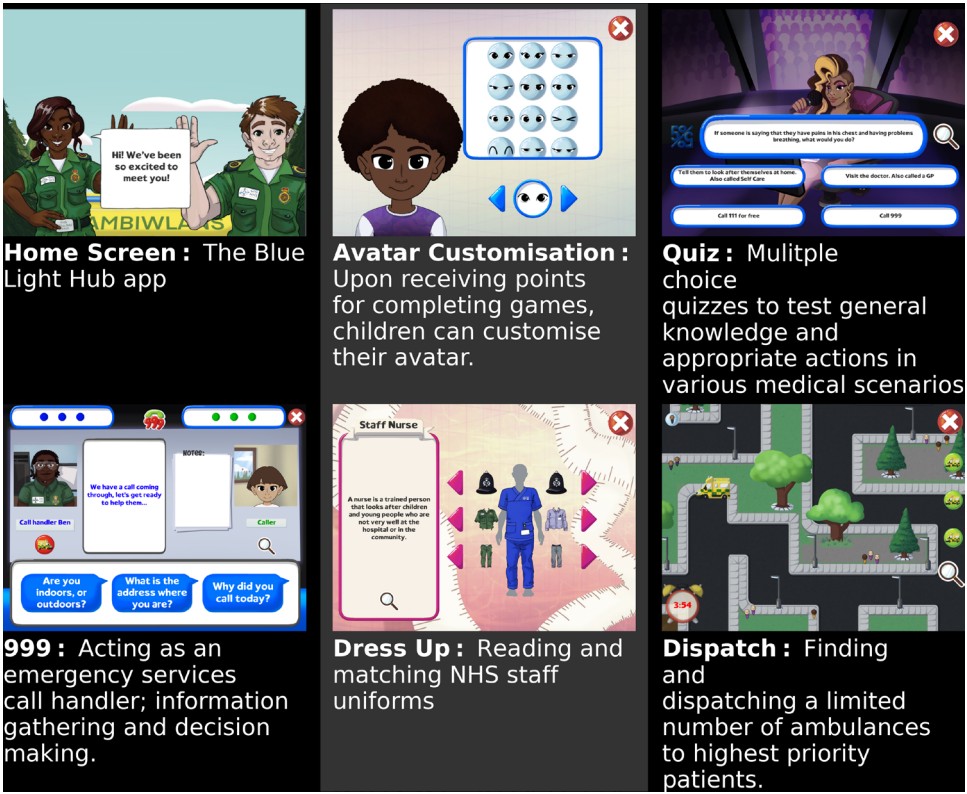

**Home Screen :** The Blue Light Hub app

**Avatar Customisation :** Upon receiving points for completing games, children can customise their avatar.

**Quiz :** Mulitple choice quizzes to test general knowledge and appropriate actions in various medical scenarios

**999 :** Acting as an emergency services call handler; information gathering and decision making.

**Dress Up :** Reading and matching NHS staff uniforms

**Dispatch :** Finding and dispatching a limited number of ambulances to highest priority patients.

**Figure 1** The Blue Light Hub app.

could withdraw from the evaluation at any time and receive minimal support when completing their quizzes (support with reading and writing only). Children completed the first quiz in class time. After completion of the first quiz, teachers were asked to use the Blue Light Hub app with children in their class for 1 hour a week for the following 2 weeks. After these 2 weeks, children completed Quiz 2. In each quiz, children were asked questions designed to assess their knowledge about emergency scenarios, their awareness of different healthcare services and their confidence in accessing support from emergency services. Nine questions were duplicated from Quiz 1 (pre-test quiz) to Quiz 2 (post-test quiz) to enable the comparison of knowledge before and after using the Blue Light Hub app. All children received certificates from the Welsh Ambulance Services University NHS Trust for taking part.

### Data analysis

Quiz responses were analysed using SPSS software V.27 (IBM). Change in knowledge, awareness and confidence before and after the use of the Blue Light Hub app was assessed using McNemar's tests, to examine differences in proportions in children's responses from pre-test to post-test quizzes. Inflated alpha levels due to multiple testing were addressed with a Bonferroni adjusted significance criterion of $p < 0.006$. A G*Power post hoc power analysis for McNemar's analysis indicated that a sample size of N=393 yielded statistical power at 0.84 at an alpha level of 0.02 for proportion of the discordant pairs 0.20, OR=2.

### Public and patient involvement statement

There was no public or patient involvement in the study design, analysis or interpretation of the results. However, pilot participants completed the quiz items to ensure they were appropriate for the age range.

### RESULTS

Table 1 shows quiz questions of interest and children's responses. For children's knowledge about emergency services, following use of the Blue Light Hub app a higher proportion of children responded that a patient should be taken to hospital in a car if they had a minor injury (hurt arm), $\chi^2(1) = 26.01$, $p < 0.001$, OR=4.38, 95% CI (2.47 to 7.77), and provided a correct question that a call handler would ask them if they called 999, $\chi^2(1) = 13.79$, $p < 0.001$ (OR=28.48, 95% CI (8.02 to 101.10); but it is worthy of note that a very high proportion of children correctly answered this question at both time points. There was no significant difference in the proportion of children who reported knowing what to do if someone had stopped breathing, $p = 0.18$. Children's follow-on free text responses to 'If yes, what would you do?' were coded for correct actions (eg, calling 999, administering cardiopulmonary resuscitation (CPR)). Similarly, there was no significant increase in children who said they knew how to help and gave a correct action, $p = 0.07$. The increase in the proportion of children who could correctly describe an emergency was also not significant, $p = 0.06$.

**Table 1** Children's quiz results before and after playing with the Blue Light Hub app

| | Quiz 1 (pre-test) | Quiz 2 (post-test) |
|---|---|---|
| **Knowledge of emergency services** | | |
| 'If someone you know hurt their arm and they needed to go to hospital, what would you do?' n (%) take in car vs call 999 | **70/360 (19.4)** | **126/374 (33.7)** |
| 'If you saw someone and they had stopped breathing, would you know what to do?' n (%) yes | 306/390 (78.5) | 319/392 (81.4) |
| 'If yes, what would you do?' n (%) correct response | 275/306 (89.9) | 299/319 (93.7) |
| 'When you ring 999 for an ambulance, we ask lots of questions. Can you think of 1 question we ask you?' n (%) correct response | **360/393 (91.6)** | **381/393 (96.9)** |
| 'You should only phone 999 for an ambulance in an 'emergency'. What do you think an 'emergency' is?' n (%) correct response | 351/393 (89.3) | 366/393 (93.1) |
| **Awareness of healthcare services** | | |
| 'If you are not feeling very well, your family and teachers can help you. Can you think of anyone else who can help you if you are not feeling very well?' n (%) reporting NHS service | **235/393 (59.8)** | **294/393 (74.8)** |
| 'There are lots of people that do different jobs in the Ambulance Service. Can you name 2 of them?' n (%) correct responses | **229/393 (58.3)** | **270/393 (68.7)** |
| 'Do you know how 111 can help you?' n (%) yes | **101/391 (25.8)** | **226/393 (57.5)** |
| **Confidence calling 999** | | |
| 'Would you feel confident if you had to ring 999 for an ambulance?' n (%) yes | 289/389 (74.3) | 298/386 (77.2) |

Note. Bold represents significant differences in proportions from the pre-test to post-test quizzes.
NHS, National Health Service.

In terms of children's awareness of healthcare services, we found that following the use of the Blue Light Hub app, there were significant increases in the proportion of children who identified an NHS service that could help them if they were not feeling well, $\chi^2(1) = 33.31$, p<0.001, OR=9.94, 95% CI (5.76 to 17.15), in children who could name two roles in the NHS, $\chi^2(1) = 12.80$, p<0.001, OR=4.35, 95% CI (2.76 to 6.84) and in children reporting that they knew how dialling 111 could help them, $\chi^2(1) = 90.05$, p<0.001, OR=3.54, 95% CI (2.09 to 5.99), (see table 1).

Finally, we did not identify a significant change in the proportion of children who reported feeling confident if they had to ring 999 for an ambulance p=0.31.

## DISCUSSION

To our knowledge, the Blue Light Hub app designed by the Welsh Ambulance Services University NHS Trust is the first of its kind to teach primary school-aged children about emergency services. Our findings suggest that using the Blue Light Hub app for just 2 hours over 2 weeks during school time was related to improvements in children's knowledge about, and awareness of, emergency services.

Importantly, following the use of the app children demonstrated a better understanding of when not to call an ambulance, suggesting the app offers opportunities to better understand actions to take in non-emergency medical scenarios. We also found that children were more likely to list NHS services that could help them if they were feeling unwell, they demonstrated a better awareness of different NHS job roles, and an increased awareness of how 111 could help them in non-emergency situations. Although we did not find significant increases in children's confidence if they were to call 999, it is noteworthy that three-quarters of children reported feeling confident and there was an increase in knowledge about what a call handler may ask them if they had to do so. Our findings extend previous research demonstrating that secondary school children tend to have high levels of knowledge about contacting emergency services[13 14]; our evaluation indicates this is also true for younger primary school-aged children. Our findings are particularly encouraging given that barriers to intervening in emergency scenarios later in childhood often involve a lack of confidence, fear of making mistakes and lack of awareness and knowledge of appropriate actions.[15]

One of the main aims of developing the Blue Light Hub app was to find novel ways to support education about urgent and emergency care to address the rise in calls for ambulance services in non-emergency scenarios. Such calls add significant pressure to an already stretched service that needs to reduce spending.[16] Research shows that reasons for ambulance use in non-emergency situations are multifaceted.[2–4] For example, recent reviews have indicated numerous factors associated with ambulance use for non-emergency problems. These included socioeconomic status, geographical location, practical reasons (ie, lack of transport), emotional reasons (ie, uncertainty, reassurance) and level of education, knowledge and experience in judging the appropriateness of ambulance use for their circumstances.[17 18] Although a small-scale intervention may not radically change the behaviour of the population, better knowledge and awareness repeatedly consolidated by the use of educational apps may encourage individuals to evaluate their options more thoroughly before seeking emergency care. Quite possibly, interventions such as the Blue Light Hub app may have the most impact in more deprived areas; future research would do well to investigate this possibility on a larger scale and explore the generalisability of this type of education-based intervention in other locations and contexts. Although this investigation focused on improving awareness, knowledge, and confidence in primary school-aged children, app-based education could also be explored as an avenue for supporting adult education about emergency services and decision-making prior to calling an ambulance.

### Improvements

The app was developed during the COVID-19 pandemic, and therefore opportunities for consultation with young people, parents, and teachers were limited. We recommend that future development of such apps involve the inclusion of both a young person and parent and teacher advisory groups alongside the lessons learnt from this initial evaluation.

In general feedback, children and teachers shared that the Blue Light Hub app was engaging and enjoyable to use. However, this initial evaluation gave insights into ways the app can be improved in terms of accessibility and improving children's learning experiences. Some teachers suggested that the app should include audio to reduce the need for reading skills and increase the accessibility of the app. Another popular suggestion by teachers was further consideration that children typically share tablet devices in the classroom and across year groups. An important step for further development of the app would be to ensure that children can access their own profiles to be able to keep track of their own progress and learning journey.

Children had many ideas for more games they would want to play on the app, including how to give CPR, first aid and more immersive experiences of being an ambulance or air ambulance driver. Indeed, one valuable direction for game development could be an experience of steps to take if a child encountered a patient who required CPR, given that evidence suggests that in later childhood, despite good knowledge of actions to take, this does not necessarily translate to practical ability.[19]

### Limitations

Although these findings are promising, there are limitations to note. This evaluation represents a first step in understanding the feasibility and effectiveness of an

educational app to support children's knowledge of emergency services, however, future evaluation of the effectiveness of the app as an intervention for children should be investigated in the context of a randomised controlled trial to evaluate the effectiveness of the app in short-term and long-term, that also accounts for the level of support that teachers provide when navigating the app. Given the short-term use of the app in this evaluation, a further avenue for investigation will be the long-term retention of knowledge and the potential need for repeated use of the app in class time. Additionally, future research would do well to understand how the app can be used to consolidate children's knowledge of emergency services alongside in-person interactive initiatives.

## CONCLUSION

In summary, our findings suggest that primary school-aged children's knowledge and awareness of emergency scenarios and services can be supported via educational apps, such as the Blue Light Hub app. Our findings corroborate research demonstrating the value of educating children on what to do in emergency scenarios and emergency care,[5] and that primary school-aged children are receptive to interventions that support their ability to recognise emergencies and call for help.[20 21] We join other calls for more emphasis on emergency response readiness in school curriculums[15 22 23]; this may have positive consequences for wider community knowledge of appropriate use of emergency services. Our findings, though preliminary, suggest that among other educational approaches, educational apps may be a useful resource for teaching.

**Contributors** AP designed the study, monitored data collection, planned the analyses, drafted and revised the paper. AP is the guarantor. FM led recruitment, monitored data collection and edited the paper. We would also like to thank Cara Houlcroft, Meg Kydd-Coutts, Maisie Bowmaster and Anna Preece for their research assistance. We would like to thank the children and teachers who took part in this study.

**Funding** AP is funded by an ESRC New Investigator Award (Grant reference: ES/T00049X/1).

**Competing interests** None declared.

**Patient and public involvement** Patients and/or the public were not involved in the design, or conduct, or reporting, or dissemination plans of this research.

**Patient consent for publication** Not applicable.

**Ethics approval** Ethical approval for the evaluation was obtained from the Cardiff University School of Psychology Ethics Committee (EC.21.12.14.6490R). Participants gave informed consent to participate in the study before taking part.

**Provenance and peer review** Not commissioned; externally peer reviewed.

**Data availability statement** Data are available upon reasonable request.

**ORCID iD**
Amy L Paine http://orcid.org/0000-0002-9025-3719

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
