## [Reviewer comments · BMJ Open]

ARTICLE DETAILS

TITLE (PROVISIONAL)	“When you're hurt and you need serious help you call 999” Educating Children about Emergency Services and Appropriate use of 999: An Evaluation Study of the Blue Light Hub App
AUTHORS	Paine, Amy; Maclean, Fiona

VERSION 1 – REVIEW

REVIEWER	Tella, Susanna LAB University of Applied Sciences - Lappeenranta kampus, Health & Wellbeing
REVIEW RETURNED	28-Aug-2023

GENERAL COMMENTS	Thank you for the revised manuscript. This looks quite clear to me. A few comments, clear research questions could be provided. In addition, the statistical analysis is described in the results. In my opinion, this could be included in data analysis.
--

REVIEWER	Proctor, Alyesha University of the West of England
REVIEW RETURNED	01-Nov-2023

GENERAL COMMENTS	Thank you for the opportunity to review this interesting manuscript on educating children about emergency services and appropriate use of 999. The Blue Light Hub App seems like a great intervention to educate young children on the use of 999. A few minor comments from me: You have not included: Proctor, A., Baxter, H. and Booker, M. (2021) What factors are associated with non-urgent problems in children? A systematic mapping review and qualitative synthesis, BMJ Open. This would be very relevant to this work and could be referred to in the discussion. The discussion section is quite limited with minimal referral to existing evidence. Needs more detail, what do these results mean and what can be done in the future with this App. Will this work outside of Wales? Link to what are the factors associated for calling 999 for non-emergency problems in children, such as socioeconomic status, parents, reason for the call, no transport. # The inclusion of both a young person's advisory group and parent advisory group (PPI) would be very beneficial.
---

REVIEWER	Chaurasia, Alok Ranjan Shyam Institute
REVIEW RETURNED	22-Jan-2024

GENERAL COMMENTS	The statical analysis of the data requires improvement. Instead of computing proportions (%) odds ratios should have been calculated through the application of logistic regression analysis for both pre-test and post-test data and controlling selected characteristics of children. For example, what is the odds ratio of taking in a car and calling 999 and how has the odds ratio changed? What are the characteristics of children that statistically significantly influence the odds ratio. The existing analysis is weak.
---

VERSION 1 – AUTHOR RESPONSE

Reviewer: 1

Thank you for the revised manuscript. This looks quite clear to me. A few comments, clear research questions could be provided. In addition, the statistical analysis is described in the results. In my opinion, this could be included in data analysis.

Thank you for your positive comment about the manuscript. We have clarified the research questions on p.2-3 and on p.4 we have included a 'Data Analysis' subheading and included details of the analysis there instead of at the beginning of the results section.

Reviewer: 2

Thank you for the opportunity to review this interesting manuscript on educating children about emergency services and appropriate use of 999. The Blue Light Hub App seems like a great intervention to educate young children on the use of 999.

Thank you so much for your positive comments on the manuscript and the Blue Light Hub App!

You have not included: Proctor, A., Baxter, H. and Booker, M. (2021) What factors are associated with non-urgent problems in children? A systematic mapping review and qualitative synthesis, BMJ Open. This would be very relevant to this work and could be referred to in the discussion.

Thank you very much for this very helpful recommendation. We have included this work and several other references to strengthen the discussion on p. 7.

The discussion section is quite limited with minimal referral to existing evidence. Needs more detail, what do these results mean and what can be done in the future with this App. Will this work outside of Wales? Link to what are the factors associated for calling 999 for non-emergency problems in children, such as socioeconomic status, parents, reason for the call, no transport. The inclusion of both a young person's advisory group and parent advisory group (PPI) would be very beneficial.

Thank you very much for your helpful comments about directions to explore in the Discussion. We have expanded the discussion section on p.7 and p.8. Firstly we have added a section to expand on the existing literature and to reflect on the factors associated with calling 999 for non-emergency problems. We have also discussed generalisability of the app and some future directions. On p. 8 we have responded to the idea of a young person's advisory and parent advisory group. We have included an 'Improvements' section to include some of the lessons we learned from administering the app for future researchers.

Reviewer: 3

The statical analysis of the data requires improvement. Instead of computing proportions (%) odds ratios should have been calculated through the application of logistic regression analysis for both pre-test and post-test data and controlling selected characteristics of children. For example, what is the odds ratio of taking in a car and calling 999 and how has the odds ratio changed? What are the characteristics of children that statistically significantly influence the odds ratio. The existing analysis is weak.

Thank you for your comments about our analytic approach. As this is very much an initial evaluation of the effectiveness of the app, the data we collected from participants was limited to the quiz results. Coupled with there being no research (to our knowledge) on child characteristics associated with decision-making in emergency and non-emergency situations, we did not have the available data nor a justification to include child characteristics as covariates in this study. As such, we realise that our analytic approach is simple, but it is the correct approach to assess pre- and post-test differences in paired nominal data (Sundjaja et al., 2023). We have thought carefully about our analytic approach and compared it to other feasibility intervention research and have found our approach to be comparable: e.g., Bennett et al., 2018 BMJ Injury and Prevention 'Feasibility of Safe-Tea: a parent-targeted intervention to prevent hold drink scalds in preschool children; Caputi et al., 2021 Frontiers in Psychology 'See & Eat!' E-books to promote vegetable eating among preschoolers. However, in response to your feedback we have now included odds-ratios which are highlighted in the Results section.

Thank you very much for all your positive comments and helpful feedback.

VERSION 2 – REVIEW

REVIEWER	Chaurasia, Alok Ranjan Shyam Institute
REVIEW RETURNED	26-Mar-2024
GENERAL COMMENTS	Statistical analysis may be elaborated.